

# *Noblella thiuni* sp. n., a new (singleton) species of minute terrestrial-breeding frog (Amphibia, Anura, Strabomantidae) from the montane forest of the Amazonian Andes of Puno, Peru

Alessandro Catenazzi[1,2] and Alex Ttito[3,4]

[1] Biological Sciences, Florida International University, Miami, FL, United States of America
[2] Centro de Ornitología y Biodiversidad, Lima, Perú
[3] Departamento de Ecología, Facultad de Ciencias Biológicas, Pontificia Universidad Católica de Chile, Santiago, Chile
[4] Museo de Historia Natural, Universidad Nacional de San Antonio Abad, Cusco, Perú

Corresponding author
Alessandro Catenazzi,
acatenaz@fiu.edu,
acatenazzi@gmail.com

## ABSTRACT

We describe a new species of minute, terrestrial-breeding frog in the genus *Noblella*. We collected a single specimen in the leaf litter of primary montane forest (2,225 m a.s.l.) near Thiuni, in the Provice of Carabaya, Department of Puno, in the upper watershed of a tributary of the Inambari River of southern Peru, the same locality where we found the types of *Psychrophrynella glauca* Catenazzi & Ttito 2018. We placed the new species within *Noblella* on the basis of molecular data, minute size, and overall morphological resemblance with the type species *N. peruviana* and other species of *Noblella*, including having three phalanges on finger IV (as in *N. coloma*, *N. heyeri*, *N. lynchi*, *N. madreselva*, *N. peruviana*, and *N. pygmaea*), and terminal phalanges T-shaped and pointed. *Noblella thiuni* sp. n. is distinguished from all other species of *Noblella* by having ventral surfaces of legs bright red, and chest and belly copper reddish with a profusion of silvery spots. The new species further differs from known Peruvian species of *Noblella* by the combination of the following characters: tympanic membrane absent, eyelids lacking tubercles, dorsal skin finely shagreen, tarsal tubercles or folds absent, three phalanges on Finger IV, tips of digits not expanded, no circumferential grooves on digits, inguinal spots present. The new species has a snout–vent length of 11.0 mm in one adult or subadult male. Our new finding confirms the high levels of endemism and beta diversity of small, terrestrial-breeding frogs inhabiting the moss layers and leaf litter in the montane forests of the Amazonian slopes of the Andes and adjacent moist puna grasslands, and suggests much work remains to be done to properly document this diversity.

## INTRODUCTION

The genus *Noblella* is distributed from Ecuador to Bolivia (Fig. 1; *Catenazzi, Uscapi & von May, 2015*), but there is uncertainty regarding the monophyly of the genus and the number of species (*Catenazzi & Ttito, 2016*; *Catenazzi & Ttito, 2018*; *De la Riva et al., 2017*). The current taxonomy recognizes 12 species of *Noblella* (*AmphibiaWeb, 2019*; *Catenazzi, Uscapi & von May, 2015*), of which four occur in southern Peru, where several undescribed species have been reported (Fig. 2; *von May et al., 2017*). The small size of these frogs (*Lehr & Catenazzi, 2009*), terrestrial life habits requiring intensive search in the leaf litter (*Catenazzi et al., 2011*), and micro-endemism of most species of high-elevation, small strabomantid frogs in the Andes (*De la Riva et al., 2017*) suggest that many additional species remain to be discovered and recognized throughout the tropical Andes (*Catenazzi, 2015*). Here we describe a new species of *Noblella* on the basis of a singleton (following terminology of *Lim, Balke & Meier, 2012*) found in the leaf litter of a cloud forest remnant in the Cordillera de Carabaya, in the southern Peruvian department of Puno, along a tributary of the Inambari River. This specimen is unlike any of the previously described species of *Noblella* or of the morphologically similar *Psychrophrynella*. We describe the species after considering the trade-off between a complex integrative approach to delimit the new species, and the need to accelerate the pace of taxonomic descriptions (*De Carvalho et al., 2008*; *Guayasamin, Arteaga & Hutter, 2018*; *Padial et al., 2009*), particularly for micro-endemic taxa inhabiting threatened cloud forests such as many species of small strabomantid frogs.

## MATERIALS & METHODS

On 14 August 2017 we (AC and AT) conducted a rapid survey (∼4 h) of the leaf litter of a relictual cloud forest along the Macusani-San Gabán road, which connects the Peruvian Altiplano to the Amazon rainforest and the interoceanic highway between Peru and Brazil. We removed the leaf litter by hand and searched opportunistically around fallen logs, rocks, moss-covered soil, etc. We searched within an area of approximately 100 m². We found a single specimen, which we euthanized with 20% benzocaine.

We wrote the diagnosis and description by following *Duellman & Lehr (2009)* and *Lynch & Duellman (1997)*, except that we used the term ''dentigerous processes of vomers'' instead of ''vomerine odontophores'' (*Duellman, Lehr & Venegas, 2006*). We follow *Heinicke et al. (2017)* and *De la Riva et al. (2017)* for family placement and taxonomy. We derived meristic traits of similar species from comparisons with museum specimens (Appendix S1), field notes of coloration in live specimens of other species, published photographs, and the original species descriptions (*Catenazzi, Uscapi & von May, 2015*). Abbreviations of collections are: CORBIDI, Herpetology Collection, Centro de Ornitología y Biodiversidad, Lima, Peru; KU, Natural History Museum, The University of Kansas, Lawrence, Kansas, USA; MHNC, Museo de Historia Natural, Universidad San Antonio Abad del Cusco, Cusco, Peru; MHNG, Muséum d'Histoire Naturelle, Genève, Switzerland; MUSM, Museo de Historia Natural, Universidad Nacional Mayor de San Marcos, Lima, Peru.

We preserved the holotype in 70% ethanol (without fixing it in formalin). We did not dissect the specimen and we are tentatively identifying it as an adult or subadult male.

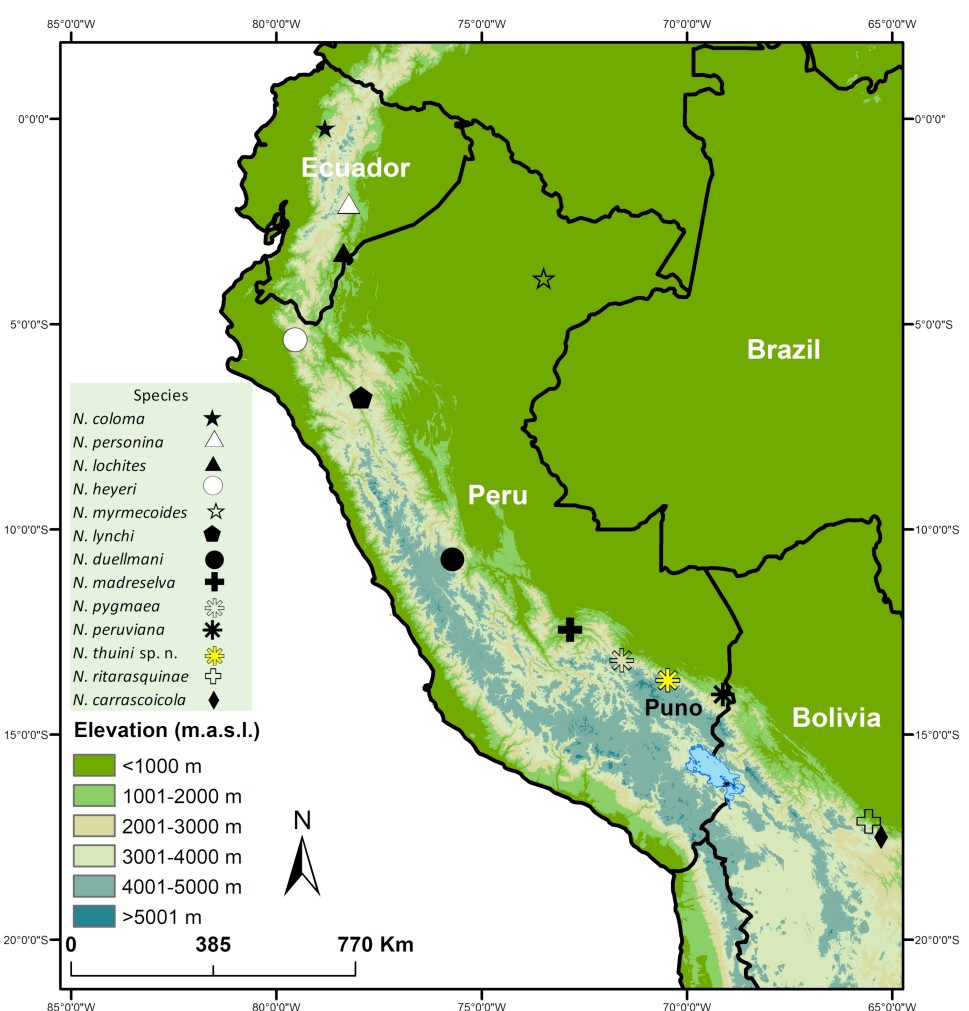

**Figure 1** **Map showing type localities of species in the genus *Noblella* (Anura, Strabomantidae).** Type localities of *N. coloma, N. lochites* and *N. personina* in Ecuador, *N. duellmani, N. heyeri, N. lynchi, N. madreselva, N. myrmecoides, N. peruviana, N. pygmaea* and *N. thiuni* sp. n. in Peru, and *N. carrascoicola* and *N. ritarasquinae* in Bolivia.

We measured the following variables to the nearest 0.1 mm with digital calipers under a stereomicroscope: snout–vent length (SVL), tibia length (TL), foot length (FL, distance from proximal margin of inner metatarsal tubercle to tip of Toe IV), head length (HL, from angle of jaw to tip of snout), head width (HW, at level of angle of jaw), eye diameter (ED), tympanum diameter (TY), interorbital distance (IOD), upper eyelid width (EW), internarial distance (IND), eye–nostril distance (E–N, straight line distance between anterior corner of orbit and posterior margin of external nares). We numbered fingers and toes preaxially to postaxially from I–IV and I–V respectively. We determined comparative lengths of toes III and V by adpressing both toes against Toe IV; lengths of fingers I and II were determined by adpressing the fingers against each other. We used field notes and photographs we took

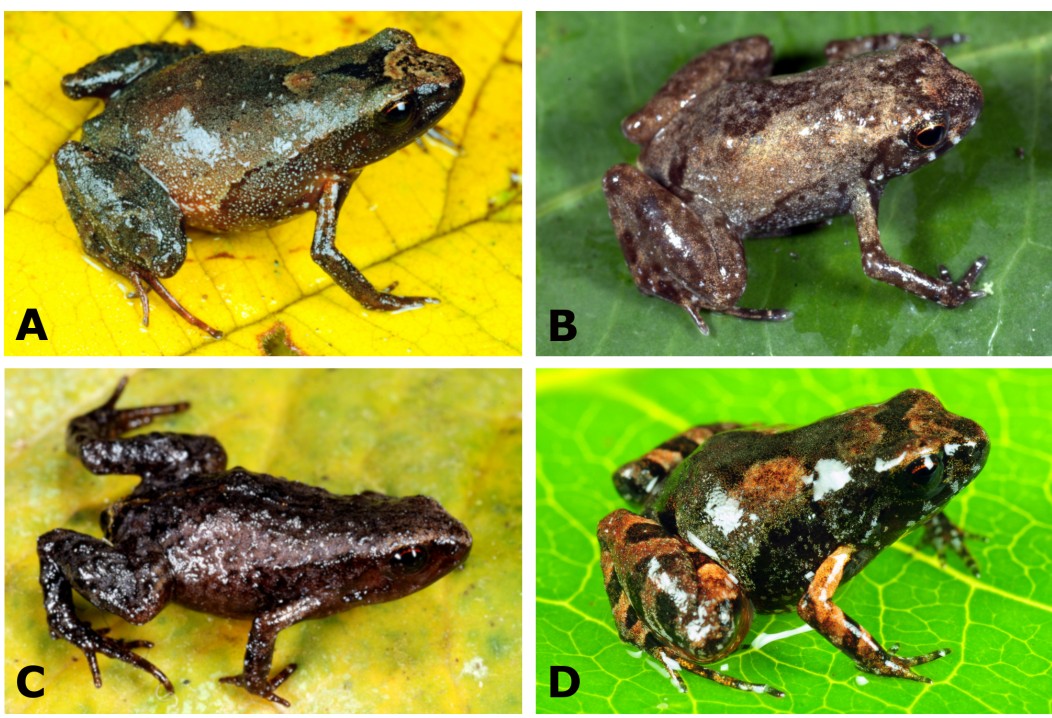

**Figure 2** **Species of *Noblella* (Anura, Strabomantidae) from southern Peru.** (A) *N. madreselva* (unvouchered female, SVL 18.9 mm) from Madre Selva, La Convención, Cusco; (B) *Noblella* sp. "SP" (male AC 95.09, SVL 13.1 mm) from San Pedro, Paucartambo, Cusco; (C) *N. pygmaea* from Wayqecha, Paucartambo, Cusco (female holotype, MUSM 26320, SVL 12.4 mm); (D) *N. thiuni* sp. n. from Thiuni, Carabaya, Puno (male holotype, CORBIDI 18723, SVL 11.0 mm). Photographs by A Catenazzi.

in the field to describe coloration in life. We have deposited photographs of the live and preserved holotype at the Calphoto online database (http://calphotos.berkeley.edu).

We used phylogenetic analyses to confirm generic placement of the new species within *Noblella* through analysis of the mitochondrial 16S rRNA fragment. This fragment is commonly used for anuran taxonomy (*Fouquet et al., 2007*; *Padial et al., 2009*; *Vences et al., 2005*), and is the sequence most commonly used for species of Holoadeninae (*Hedges, Duellman & Heinicke, 2008*). We used liver tissues from the holotype of *N. thiuni* sp. n. to obtain DNA sequences for the new species (accession code MK072732; Appendix S2). We also obtained DNA sequences from species of *Noblella* (*Catenazzi, Uscapi & von May, 2015*) and *Psychrophrynella* (*Catenazzi & Ttito, 2016*; *Catenazzi & Ttito, 2018*; *von May et al., 2017*), and downloaded sequences of closely related genera within Holoadenindae (*Barycholos*, *Bryophryne*, *Holoaden*, and *Microkayla*) from GenBank (Appendix S2). Our phylogenetic analyses are preliminary, because there is uncertainty concerning the taxonomic position of *Noblella* and *Psychrophrynella*, and because genetic sequences of their type species (*N. peruviana* and *P. bagrecito*) are not available (*Catenazzi & Ttito, 2018*). We extracted DNA from liver or skin swab samples (*N. madreselva*) with a commercial extraction kit (IBI Scientific, Peosta, USA). We followed *Hedges, Duellman & Heinicke (2008)* for DNA amplification and sequencing, and used the 16Sar (forward) primer

(5′-3′sequence: CGCCTGTTTATCAAAAACAT) and the 16Sbr (reverse) primer (5′-3′sequence: CCGGTCTGAACTCAGATCACGT). The thermocycling conditions during the polymerase chain reaction (PCR) were: one cycle at 96 °C/3 min; 35 cycles at 95 °C/30 s, 55 °C/45 s, 72 °C/1.5 min; and one cycle at 72 °C/7 min. We used a Proflex thermal cycler (Applied Biosystems), purified PCR products with Exosap-IT (ThermoFisher), and shipped purified samples to MCLAB (South San Francisco, CA, USA) for sequencing. We used Geneious, version 11.1.5 (Biomatters, http://www.geneious.com/) to align sequences with the MAFFT v7.017 alignment program (*Katoh & Standley, 2013*), and trimmed sequences to a length of 537 bp. Our analysis included 22 terminals. We employed a Maximum Likelihood (ML) approach to infer a molecular phylogeny using MEGA v. 7 (*Kumar, Stecher & Tamura, 2016*) based on the General Time Reversible model. Initial tree(s) for the heuristic search were obtained by applying Neighbor-Join and BioNJ algorithms to a matrix of pairwise distances estimated using the Maximum Composite Likelihood (MCL) approach, and then selecting the topology with superior log likelihood value. A discrete Gamma distribution was used to model evolutionary rate differences among sites (5 categories (+G, parameter = 0.7550)). The rate variation model allowed for some sites to be evolutionarily invariable ([+I], 41.29% sites). We assessed node support using 500 bootstrap replicates. We also estimated pairwise, uncorrected genetic distances (p-distances) for 16S rRNA between the new species and other species of *Noblella* and *Psychrophrynella*, as well as species from other genera of Holoadeninae.

Our research was approved by the Institutional Animal Care and Use Committee of Southern Illinois University Carbondale (protocol #16-006) and Florida International University (protocol #18-009). The Dirección General Forestal y de Fauna Silvestre, Ministerio de Agricultura y Riego issued the permit authorizing this research (permits # 0292-2014-MINAGRI-DGFFS/DGEFFS, #029-2016-SERFOR-DGSPFS).

The electronic version of this article in Portable Document Format (PDF) will represent a published work according to the International Commission on Zoological Nomenclature (ICZ), and hence the new name contained in the electronic version is effectively published under that Code from the electronic edition alone. This published work and the nomenclatural acts it contains have been registered in ZooBank, the online registration system for the ICZN. The ZooBank LSIDs (Life Science Identifiers) can be resolved and the associated information viewed through any standard web browser by appending the LSID to the prefix http://zoobank.org/. The LSID for this publication is: urn:lsid:zoobank.org:pub:3F917F8B-5D31-44D4-86E2-389341A21BD1. The online version of this work is archived and available from the following digital repositories: PeerJ, PubMed Central and CLOCKSS.

## RESULTS

### Generic placement

Our phylogenetic analysis indicates relatedness of our specimen with species of *Noblella* from southern Peru (Fig. 3), but also with species of *Psychrophrynella*. As previously discussed (*Catenazzi & Ttito, 2016*; *Catenazzi & Ttito, 2018*; *De la Riva et al., 2017*; *De la*

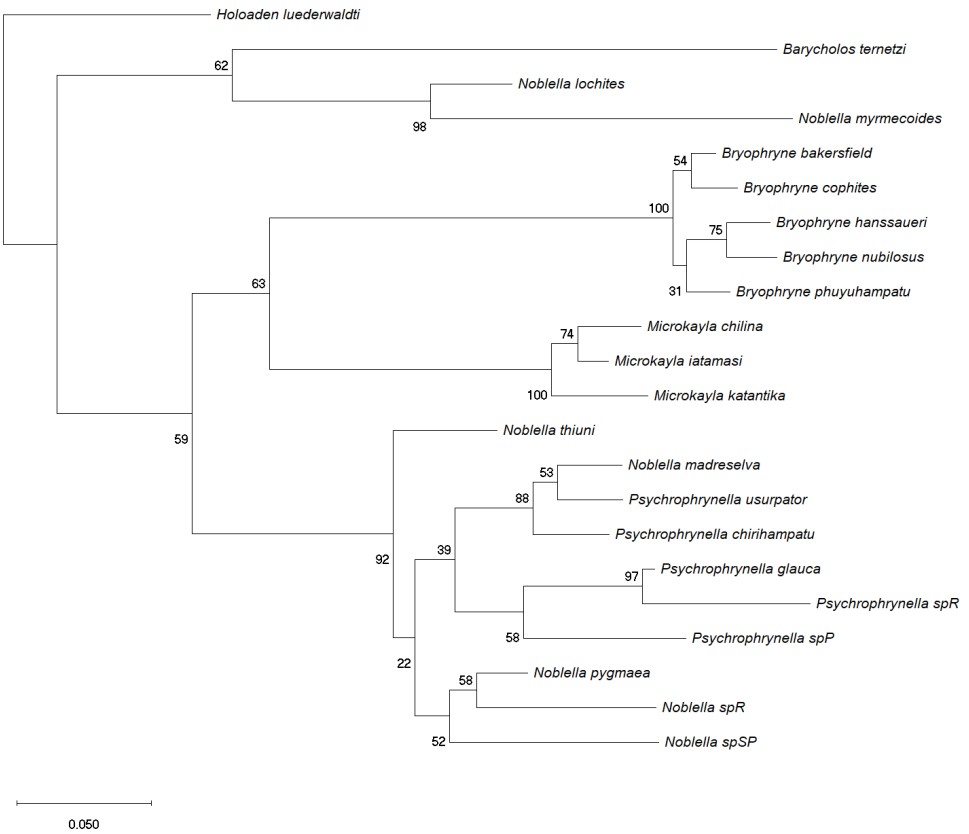

**Figure 3** **Phylogenetic analysis of 16S rRNA by using Maximum Likelihood.** Maximum likelihood optimal tree with bootrsap node values from the analysis of a dataset of 537 bp of 22 species aligned by MAFFT based on the General Time Reversible model. Initial tree(s) for the heuristic search were obtained by applying Neighbor-Join and BioNJ algorithms to a matrix of pairwise distances estimated using the Maximum Composite Likelihood (MCL) approach, and then selecting the topology with superior log likelihood value. A discrete Gamma distribution was used to model evolutionary rate differences among sites (5 categories (+$G$, parameter = 0.7550)). The rate variation model allowed for some sites to be evolutionarily invariable ([+$I$], 41.29% sites). The tree is drawn to scale, with branch lengths measured in the number of substitutions per site. All positions containing gaps and missing data were eliminated.

Riva, Chaparro & Padial, 2008), the type species of both *Noblella* and *Psychrophrynella*, *N. peruviana* and *P. bagrecito*, respectively, have not been included in phylogenetic analyses due to lack of DNA sequences. Similarity in meristic traits (i.e., presence of tarsal folds or tubercles), and our present phylogeny showing *N. thiuni* sp. nov. as the sister taxon to all other species of sampled *Noblella* and *Psychrophrynella*, supports the idea that *Psychrophrynella* species may be nested within a larger *Noblella* clade. However, the taxonomy cannot be resolved without collecting more information from the type species *N. peruviana* and *P. bagrecito*, including DNA sequences, morphometric measurements, and recordings of advertisement calls. Our phylogeny produced from a small dataset of only one mitrochondrial gene fragment (16S) is similar to previously published trees generated using much larger datasets (i.e., De la Riva et al., 2017; Hedges, Duellman & Heinicke, 2008). Our tree however includes a larger number of *Noblella* species, including

undescribed forms from southern Peru (*von May et al., 2017*). These southern Peruvian species, which occur close to the type locality of *N. peruviana* and *P. bagrecito* in the Departments of Puno and Cusco, form a sister clade to *Microkayla* and *Bryophryne*, and thus appear not to be closely related to species of *Noblella* from northern Peru and Ecuador.

The new species is assigned to *Noblella*, as defined by *De la Riva, Chaparro & Padial (2008)*, *Duellman & Lehr (2009)*, and *Hedges, Duellman & Heinicke (2008)*, on the basis of the frog's minute size and overall morphological resemblance with the type species *Noblella peruviana* and other species of *Noblella*, including having three phalanges on Finger IV (as in *N. coloma*, *N. heyeri*, *N. lynchi*, *N. madreselva*, *N. peruviana*, and *N. pygmaea*), terminal phalanges T-shaped and pointed, tympanic membrane absent (as in *N. duellmani, N. peruviana*). The species with the lowest uncorrected p-distance for 16S rRNA (Table 1) is *N. pygmaea* (0.07), followed by *N. madreselva* (0.08) and *N.* sp. R (*von May et al., 2017*) and four species of *Psychrophrynella* (0.09). Frogs of the genus *Noblella* are morphologically similar and closely related to *Barycholos, Bryophryne, Holoaden, Microkayla* and *Psychrophrynella* (*De la Riva et al., 2017*; *Hedges, Duellman & Heinicke, 2008*; *Heinicke, Duellman & Hedges, 2007*; *Padial, Grant & Frost, 2014*). The new species is assigned to *Noblella* rather than *Barycholos* (characters in parentheses), because it lacks the dentigerous processes of the vomers (present), has Finger I shorter than Finger II (Finger I >Finger II), and has low, rounded subarticular tubercles (subarticular tubercles elevated). The new species differs from species of *Bryophryne* (characters in parentheses) in having T-shaped terminal phalanges (knob-shaped), no nuptial pads (present or absent), tarsal fold (absent), small size and slender body with longer limbs (larger size with stubby body and short limbs). Species in the genus *Holoaden* have prominent dentigerous processes of vomers (absent in *N. thiuni* sp. n.), terminal phalanges knob-shaped (T-shaped), venter areolate (smooth), and much larger size (up to 48 mm SVL) than species of *Noblella*. The new species differs from species of *Microkayla* (characters in parentheses) in having T-shaped terminal phalanges (absent), Toe V about the same length as Toe III (Toe V slightly longer than Toe III), elongated tongue (rounded), smooth venter (areolate), tarsal fold (absent), and small size and slender body with longer limbs (larger size with robust body and short extremities). There presently are no meristic traits to differentiate species of *Noblella* from species of *Psychrophrynella*, and future work will aim at resolving the taxonomic conundrum posed by the absence of synapomorphies for these two genera.

*Noblella thiuni* **sp. n.** urn:lsid:zoobank.org:pub:3F917F8B-5D31-44D4-86E2-389341A21BD1; urn:lsid:zoobank.org:act:1B9D35DE-0BA2-44E1-9559-62E30D239A30.

## Holotype

CORBIDI 18723, an adult male from 13.67603S; 70.46588W (WGS84), 2225 m a.s.l., near Thiuni, Distrito Ollachea, Provincia Carabaya, Departamento Puno, Peru, collected by A. Catenazzi and A. Ttito on 14 August 2017 (Figs. 1–3).

## Characterization

A species of *Noblella* characterized by (1) skin on dorsum finely shagreen; skin on venter smooth, discoidal fold not visible, thin dorsolateral folds visible on anterior half part of

**Table 1** **Pairwise uncorrected p-distance for 16S rRNA between *Noblella thiuni* sp. n. and related taxa in the subfamily Holadeninae.** Lowest genetic distances for *N. thiuni* are in bold. Specimen codes are listed in Appendix S2.

| | Barycholos ternetzi | Bryophryne bakersfield | Bryophryne cophites | Bryophryne hanssaueri | Bryophryne nubilosus | Bryophryne phuyuhampatu | Holoaden luederwaldti | Microkayla chilina | Microkayla iatamasi | Microkayla katantika | Noblella lochites | Noblella madreselva | Noblella myrmecoides | Noblella pygmaea | Noblella spR | Noblella spSP | Noblella thiuni | P. chirihampatu | P. glauca | P.a spP | P. spR | P. usurpator |
|---|---|---|---|---|---|---|---|---|---|---|---|---|---|---|---|---|---|---|---|---|---|---|
| Barycholos ternetzi | | | | | | | | | | | | | | | | | | | | | | |
| Bryophryne bakersfield | 0.19 | | | | | | | | | | | | | | | | | | | | | |
| Bryophryne cophites | 0.19 | 0.02 | | | | | | | | | | | | | | | | | | | | |
| Bryophryne hanssaueri | 0.21 | 0.04 | 0.03 | | | | | | | | | | | | | | | | | | | |
| Bryophryne nubilosus | 0.21 | 0.04 | 0.05 | 0.03 | | | | | | | | | | | | | | | | | | |
| Bryophryne phuyuhampatu | 0.20 | 0.03 | 0.04 | 0.04 | 0.04 | | | | | | | | | | | | | | | | | |
| Holoaden luederwaldti | 0.18 | 0.16 | 0.16 | 0.16 | 0.16 | 0.16 | | | | | | | | | | | | | | | | |
| Microkayla chilina | 0.19 | 0.14 | 0.14 | 0.15 | 0.15 | 0.13 | 0.14 | | | | | | | | | | | | | | | |
| Microkayla iatamasi | 0.19 | 0.13 | 0.13 | 0.15 | 0.15 | 0.13 | 0.15 | 0.03 | | | | | | | | | | | | | | |
| Microkayla katantika | 0.19 | 0.15 | 0.15 | 0.16 | 0.15 | 0.14 | 0.15 | 0.05 | 0.04 | | | | | | | | | | | | | |
| Noblella lochites | 0.16 | 0.19 | 0.19 | 0.19 | 0.20 | 0.18 | 0.13 | 0.18 | 0.18 | 0.17 | | | | | | | | | | | | |
| Noblella madreselva | 0.17 | 0.18 | 0.16 | 0.17 | 0.18 | 0.18 | 0.15 | 0.14 | 0.13 | 0.14 | 0.17 | | | | | | | | | | | |
| Noblella myrmecoides | 0.18 | 0.21 | 0.21 | 0.23 | 0.22 | 0.21 | 0.18 | 0.20 | 0.21 | 0.19 | 0.11 | 0.21 | | | | | | | | | | |
| Noblella pygmaea | 0.18 | 0.17 | 0.15 | 0.16 | 0.17 | 0.16 | 0.14 | 0.13 | 0.14 | 0.14 | 0.17 | 0.08 | 0.19 | | | | | | | | | |
| Noblella spR | 0.20 | 0.17 | 0.16 | 0.16 | 0.18 | 0.18 | 0.14 | 0.15 | 0.15 | 0.15 | 0.17 | 0.09 | 0.19 | 0.07 | | | | | | | | |
| Noblella spSP | 0.18 | 0.19 | 0.17 | 0.18 | 0.19 | 0.19 | 0.16 | 0.15 | 0.15 | 0.15 | 0.16 | 0.09 | 0.20 | 0.08 | 0.10 | | | | | | | |
| **Noblella thiuni** sp. n. | 0.16 | 0.15 | 0.15 | 0.14 | 0.15 | 0.15 | 0.15 | 0.12 | 0.12 | 0.14 | 0.18 | **0.08** | 0.18 | **0.07** | **0.09** | 0.10 | | | | | | |
| P. chirihampatu | 0.16 | 0.18 | 0.17 | 0.18 | 0.19 | 0.18 | 0.17 | 0.15 | 0.14 | 0.13 | 0.17 | 0.05 | 0.19 | 0.08 | 0.11 | 0.10 | **0.09** | | | | | |
| P. glauca | 0.17 | 0.18 | 0.17 | 0.18 | 0.19 | 0.18 | 0.18 | 0.14 | 0.13 | 0.13 | 0.18 | 0.09 | 0.20 | 0.10 | 0.13 | 0.12 | **0.09** | 0.09 | | | | |
| P. spP | 0.19 | 0.19 | 0.17 | 0.18 | 0.19 | 0.19 | 0.16 | 0.16 | 0.16 | 0.15 | 0.19 | 0.10 | 0.19 | 0.09 | 0.10 | 0.11 | **0.09** | 0.10 | 0.09 | | | |
| P. spR | 0.17 | 0.20 | 0.19 | 0.18 | 0.18 | 0.18 | 0.19 | 0.17 | 0.16 | 0.16 | 0.20 | 0.10 | 0.20 | 0.12 | 0.15 | 0.14 | 0.11 | 0.11 | 0.06 | 0.11 | | |
| P. usurpator | 0.17 | 0.19 | 0.18 | 0.17 | 0.18 | 0.19 | 0.15 | 0.16 | 0.15 | 0.14 | 0.17 | 0.04 | 0.22 | 0.09 | 0.11 | 0.10 | **0.09** | 0.05 | 0.09 | 0.10 | 0.10 | |

body; (2) tympanic membrane not differentiated, tympanic annulus barely visible below skin; (3) snout short, bluntly rounded in dorsal view and in profile; (4) upper eyelid lacking tubercles, narrower than IOD; cranial crests absent; (5) dentigerous process of vomers absent; (6) vocal slits present; nuptial pads absent; (7) Finger I shorter than Finger II; tips of digits bulbous, not expanded laterally; (8) fingers lacking lateral fringes; (9) ulnar tubercles absent; (10) heel lacking tubercles; inner edge of tarsus bearing an elongate, obliquous fold-like tubercle; (11) inner metatarsal tubercle elliptical and about the same size of ovoid, outer metatarsal tubercle; supernumerary plantar tubercles absent; (12) toes lacking lateral fringes; webbing absent; Toe V about the same length as Toe III; tips of digits not expanded, weakly pointed; (13) dorsum tan with a dark brown X-shaped middorsal mark and dark brown markings; interorbital bar present; chest and belly copper reddish with a profusion of silvery spots; ventral surfaces of legs bright red; throat and palmar and plantar surfaces brown; (14) SVL 11.0 mm in one male.

## Diagnosis

*Noblella thiuni* is most similar to *N. peruviana* but differs from this and other known species in the genus (*Catenazzi, Uscapi & von May, 2015*) by having ventral surfaces of legs bright red, and chest and belly copper reddish with a profusion of silvery spots (Fig. 4). *Noblella thiuni* has three phalanges on Finger IV and differs from *N. carrascoicola*, *N. lochites*, *N. myrmecoides*, and *N. ritarasquinae* which have two phalanges on Finger IV (*De la Riva & Köhler, 1998*; *Duellman & Lehr, 2009*; *Guayasamin & Teran-Valdez, 2009*; *Harvey et al., 2013*; *Köhler, 2000*). It further differs from *N. myrmecoides* in having tips of toes not expanded (slightly expanded in *N. duellmani*, *N. heyeri*, and *N. lynchi*, tips of toes teardrop-shaped in *N. myrmecoides*) and in lacking circumferential grooves (present in *N. myrmecoides*). Among the eight species with three phalanges on Finger IV, it differs from *N. heyeri*, *N. lynchi* and *N. pygmaea* by lacking a distinct tympanic membrane, and from *N. coloma*, *N. duellmani*, *N. madreselva* and *N. personina* by having a tarsal fold. It further differs from *N. madreselva* by lacking a large white mark on venter, from *N. duellmani*, *N. personina*, and *N. pygmaea* by having inguinal spots, from all species but *N. coloma*, *N. lochites*, *N. myrmecoides* and *N. peruviana* in having a finely shagreen dorsum, and from all species but *N. heyeri*, *N. lynchi*, *N. myrmecoides* and *N. peruviana* by lacking tubercles on eyelids. Finally, *N. thiuni* differs from *N. peruviana* by having a smaller inner metatarsal tubercle, about the same size as outer metatarsal tubercle (inner tubercle large, about twice the size of outer metatarsal tubercle), Toe V about the same length of Toe III (Toe V shorter than Toe III).

The new species is similar to species of *Psychrophrynella*, including the sympatric *P. glauca* (*Catenazzi & Ttito, 2018*) and the type species *P. bagrecito* (*Lynch & Duellman, 1997*), which both have a fold-like tubercle on the inner edge of tarsus, small size reaching ~19 mm, and a prominent ovoid outer metatarsal tubercle (of same or larger size than inner metatarsal tubercle). Furthermore, *N. thiuni* shares with *P. glauca* the red coloration on ventral surfaces of legs and venter, and the profusion of silvery or bluish-gray flecks on ventral surfaces of head, body, and legs. *Noblella thiuni* differs from *P. bagrecito* in having red coloration extending to all ventral surfaces including chest and throat, relative finger lengths when

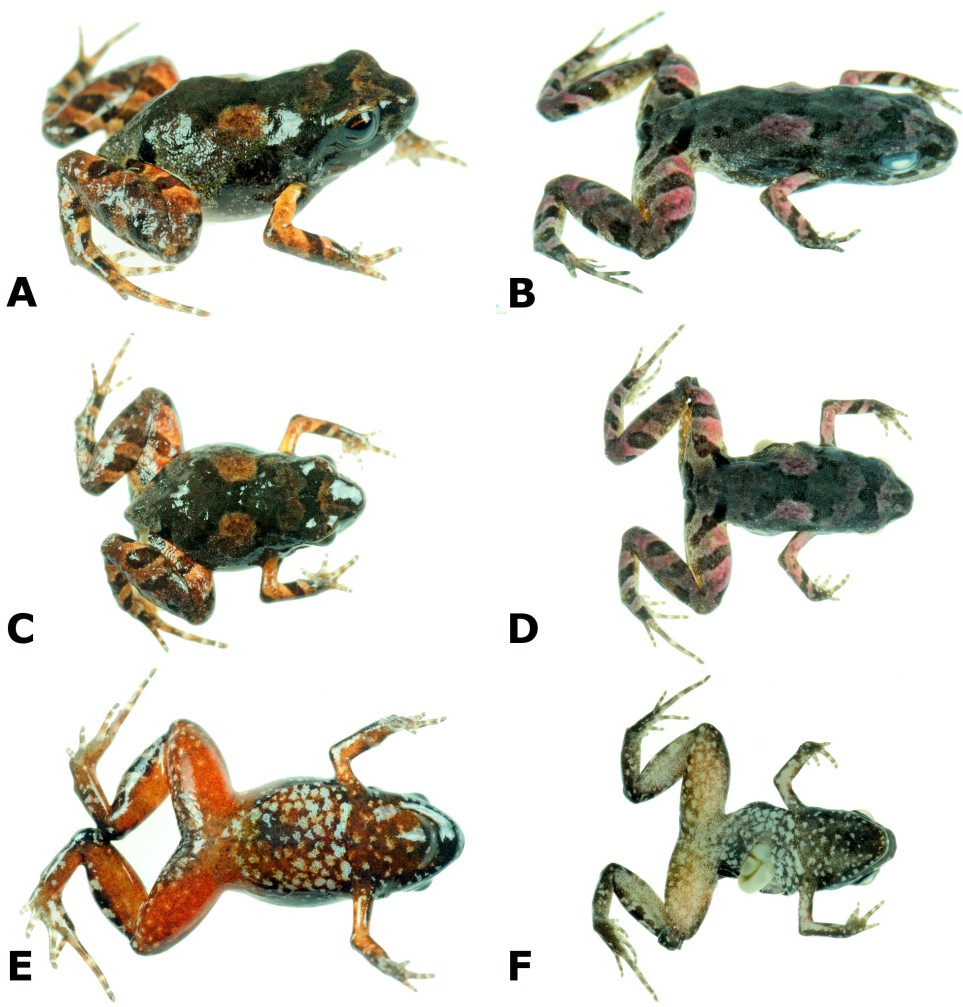

**Figure 4** **Photographs of live and preserved specimen of the holotype of *Noblella thiuni* sp. n** Live (A, C, E) and preserved (B, D, F) holotype of *Noblella thiuni* sp. n., male CORBIDI 18723 (SVL 11.0 mm) in dorsolateral (A, B), dorsal (C, D) and ventral (E, F) views. Photographs by A Catenazzi.

adpressed, Finger 3 > 2 > 4 > 1 (3 > 4 > 2 > 1 in *P. glauca*), proportionally shorter head, and presumably in having smaller size. When recently preserved, *N. thiuni* clearly stands out from specimens of *P. glauca* due to reddish marks turning purple on dorsum and dorsal surfaces of legs; the purple coloration fades to brown after several months of preservation. *Noblella thiuni* can be distinguished from *P. bagrecito* (characters in parenthesis for *P. bagrecito*) by having the snout short and bluntly rounded (snout moderately long, rounded in dorsal view and in profile), skin on venter smooth (areolate), dorsal coloration with broad markings and purple marks middorsally and on legs (longitudinal stripes, no red marks), and ventral coloration in preservative brown with silvery flecks (white to cream with brown mottling). The other two species of *Psychrophrynella*, *P. chirihampatu* and *P. usurpator*, are much larger in size (reaching 27.7 and 30.5 mm SVL, respectively), and have

ventral coloration yellow with reddish-brown or gray flecks, and dull brown, gray or black with cream flecks, respectively.

**Description of holotype.** Adult or subadult male (11.0 mm SVL); head narrower than body, its length 31% of SVL; head slightly longer than wide; head width 32% of SVL; snout very short, bluntly rounded in dorsal and lateral view (Fig. 4), eye large, 44% of head length, its diameter 1.87 times as large as its distance from the nostril; nostrils not protuberant, situated close to snout; canthus rostralis slightly curved in dorsal view, rounded in profile; lores flat; lips rounded; upper eyelids lacking tubercles; upper eyelid width 69% of interorbital distance; supratympanic fold short; tympanic membrane absent, tympanic annulus not visible; one long, postrictal ridges barely visible. Choanae minute, round, positioned anteriorly and laterally, widely separated from each other, slightly concealed by palatal shelf of maxilla; dentigerous processes of vomer and vomerine teeth absent; tongue long and narrow, about three times as long as wide.

Skin on dorsum finely shagreen, lacking tubercles; thin dorsolateral folds visible on anterior half part of body; skin on flanks smooth; skin on ventral surfaces and gular regions smooth; pectoral fold present, discoidal fold not evident; cloaca protuberant; cloacal region bearing several small tubercles. Palmar tubercle prominent, oval, approximately 3x the size of elongate, thenar tubercle; low supernumerary palmar tubercles present; subarticular tubercles prominent, ovoid in ventral view, rounded in lateral view, largest at base of fingers; fingers lacking lateral fringes; Finger IV has three phalanges; when adpressed, Finger 3 > 2 > 4 > 1 (Fig. 3); finger tips rounded, circumferential grooves absent (Fig. 3); forearm lacking tubercles.

Hindlimb lengths moderate, tibia length 53% of SVL; foot length 50% of SVL; upper and posterior surfaces of hindlimbs finely shagreen to pustulate; heels lacking tubercles; outer surface of tarsus without tubercles; inner metatarsal tubercle, oval, about the same size of conical, rounded outer metatarsal tubercle; low plantar supernumerary tubercles present; subarticular tubercles rounded, ovoid in dorsal view; toes with narrow lateral fringes, basal webbing absent; toe tips slightly acuminate, circumferential grooves absent; digital tip of Toe V smaller than tips of Toes III—IV; when adpressed, relative lengths of toes: 4 > 3 > 5 > 2 > 1 (Fig. 5).

Measurements of holotype (in mm): SVL 11.0, TL 5.8, FL 5.5, HL 3.4, HW 3.5, ED 1.5, IOD 1.6, EW 1.1, IND 0.8, E–N 0.8. Proportions as follows: TL/SVL 0.53, FL/SVL 0.50, HL/SVL 0.31, HW/SVL 0.32, HW/HL 1.03, E-N/ED 0.53, EW/IOD 0.69.

**Coloration of holotype in alcohol.** Dorsal surfaces of head and body tan with a dark brown X-shaped middorsal mark and a dark interorbital bar (not bordered by cream stripe). In the first weeks after preservation, the holotype had red coloration at the center of the X middorsal mark, and on dorsal surfaces of all limbs (Fig. 4); this red coloration faded to grayish tan after several months of preservation. Dorsal surfaces of forearms and hindlimbs with transverse dark bars. There are supralabial dark marks separated by transverse cream stripes. The iris is dark gray. Flanks dark tan anteriorly, grayish tan posteriorly. Suprainguinal marks distinct, elongated, reaching the inguinal region. Pericloacal region dark tan, contrasting from grayish tan dorsal coloration anteriorly, fading to yellowish brown coloration with cream marks posterior surfaces of thighs. The throat has dark brown

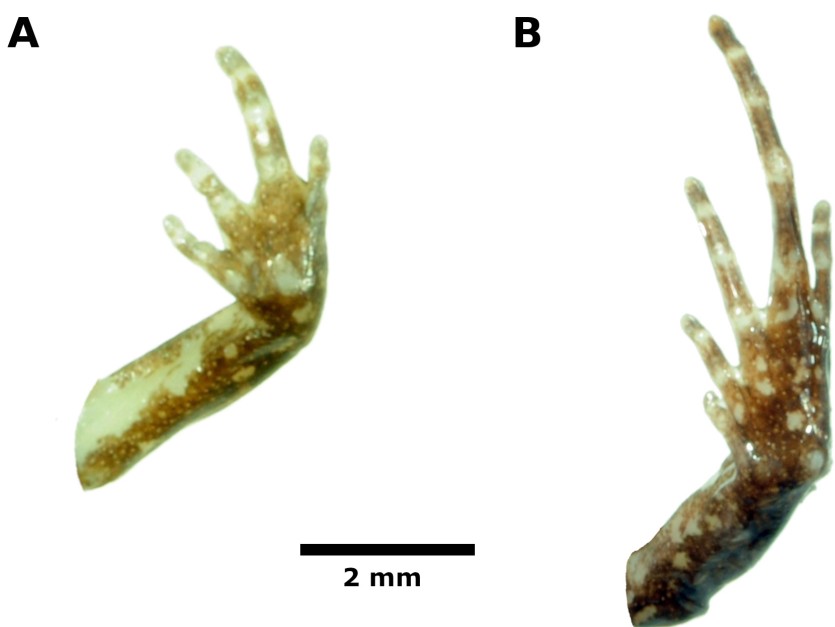

**A**

**B**

2 mm

**Figure 5** **Palmar and plantar surfaces of the holotype of *Noblella thiuni* sp. n** Ventral views of hand (**A**) and foot (**B**) of holotype, CORBIDI 18723 (hand length 2.2 mm, foot length 5.5 mm) of *Noblella thiuni* sp. n. Photographs by A Catenazzi.

coloration anteriorly, fading into pale grey with cream flecks on posteriorly. Chest and belly grayish tan with numerous cream marks, larger in diameter than flecks on throat. Ventral surface of thighs yellowish brown with cream marks and flecks; ventral surfaces of lower legs and front limbs grayish with darker spots. Plantar and palmar surfaces are brown, but fingers and toes are cream.

**Coloration of holotype in life.** Overall coloration in life similar to coloration in preservative, except that all ventral surfaces and especially the ventral surface of hindlimbs are suffused with bright red, ventral flecks and marks are silver-bluish, fingers and toes have red marks, including at the tip, and dorsal regions that turned purple red the first week after preservation (around middorsal X mark, and dorsal surfaces of limbs) are beige, and iris is dark tan with bronze reticulations and lining.

**Etymology.** The name of the new species refers to the type locality and only known locality in Thiuni, Department of Puno, Peru.

**Distribution, natural history and threats.** The cloud forest at the type locality covers a ridge that separates two creeks and is accessible through a power line maintenance trail. We found *N. thiuni* in the leaf litter along with four specimens of *Psychrophrynella glauca*, and other frog species (*Gastrotheca testudinea*, *Pristimantis platydactylus*, and an unnamed *Pristimantis* sp.) in the understory vegetation or within bromeliads (*Catenazzi & Ttito, 2018*). Although much of the Ollachea Valley has been deforested, some relictual forests remain on steep slopes and ridges, and may support more populations of these frogs and of other unreported amphibians. In absence of more details concerning the distribution and population abundance of *N. thiuni*, and despite the known threats of deforestation,

agriculture and hydropower development in the region (*Catenazzi, Lehr & Vredenburg, 2014*; *Catenazzi & von May, 2014*), we suggest an IUCN Red List threat assessment category of Data Deficient (*IUCN, 2013*). We recommend to include distribution and population surveys for this species, and the sympatric *Psychrophrynella glauca*, in any environmental assessment associated with hydroelectric development, power line maintenance, road construction, and similar large scale projects affecting the upper Ollachea Valley.

## DISCUSSION

The leaf litter of Andean cloud forests is likely to support a large undescribed diversity of small terrestrial-breeding frogs. Previous studies have documented the high levels of endemism and beta diversity of terrestrial-breeding frogs in moist grasslands of the eastern side of the Andes in southern Peru and Bolivia (*Catenazzi & Ttito, 2016*; *De la Riva et al., 2017*; *Lehr & Catenazzi, 2010*). The number of highly endemic species could be even higher in the leaf litter of cloud forests, montane scrub and other habitats below the treeline (~3,200–3,600 m in southern Peru). For example, several species of small terrestrial-breeding frogs inhabit the montane forest along the gradient near Manu National Park, occupying the elevational range from the Amazon lowland to the treeline at 3,600 m a.s.l. (*Catenazzi, Lehr & Vredenburg, 2014*; *von May et al., 2017*). Our short survey at the type locality of *Noblella thiuni* led to the discovery of another related species, *Psychrophrynella glauca* (*Catenazzi & Ttito, 2018*). It is likely that additional species of leaf litter terrestrial-breeding frogs will be discovered from the Cordillera de Carabaya and other mountain ranges in southern Peru. Discovering additional species will require targeted searches in the leaf litter, for example by using leaf litter plots (*Catenazzi et al., 2011*; *Lehr & Catenazzi, 2009*).

Our description is based on a singleton (following the terminology of *Lim, Balke & Meier, 2012*) and thus does not include information concerning sexual dimorphism and variation in morphological traits or genetic sequences. As discussed by previous authors, there is a trade-off between the use of integrative taxonomy to delimit new species and enhance taxonomic stability, and the need to accelerate the pace of taxonomic descriptions (*Carvalho et al., 2008*; *Guayasamin, Arteaga & Hutter, 2018*; *Padial et al., 2010*). *Guayasamin, Arteaga & Hutter (2018)* propose four recommendations for descriptions based on singletons or small type series. These recommendations include the need for a diagnosis relying on traits that present low intraspecific variation among similar genera, congruence among different data sets supporting the validity of the new taxon, use of well-preserved specimens, and precise locality data of type material.

In the case of *N. thiuni*, diagnostic traits include presence and relative size of tubercles and toe lengths, which are commonly used diagnostic features in this group. These traits differentiate *N. thiuni* from the most similar species, *N. peruviana*. We have precise locality data concerning *N. thiuni*, and a description of the type locality of *N. peruviana* as "the vicinity of the Inca Mine in Santo Domingo", Provincia Carabaya (the type locality was incorrectly reported in the original description; see *De la Riva, Chaparro & Padial, 2008*). Santo Domingo is a cloud forest at 1,690 m asl in the upper watershed of a small tributary

of the upper Inambari River, ~150 km SE of the type locality of *N. thiuni*. Furthermore, the type locality of *N. thiuni* is at higher elevation (2,225 m a.s.l.) in the Ollachea Valley, a tributary of the lower Inambari River. In light of the high endemism of small strabomantid frogs in southern Peru and Bolivia, and the small elevational ranges of many species of *Noblella* and *Psychrophrynella* (*De la Riva et al., 2017*; *von May et al., 2017*), the large geographic distance, difference in habitat types and elevations between *N. thiuni* and *N. peruviana* are congruent with the hypothesis that the two species are different. Furthermore, the threat of deforestation and hydroelectric development at the type locality of *N. thiuni* further encouraged our decision to name the new taxon. The first step of any conservation actions is recognizing the unicity of the organisms and their habitats. We cannot list unnamed species as threatened. The presence of threatened species often triggers the need for conservation actions aimed at preventing or mitigating changes in land use that endanger their populations.

Our study does not resolve the taxonomic uncertainty regarding the relationships among species of *Noblella* and *Psychrophrynella* (*Catenazzi & Ttito, 2018*; *De la Riva et al., 2017*; *De la Riva, Chaparro & Padial, 2008*). There are no known synapomorphies for these two morphologically similar genera, and there is little information beyond external morphology available for most species. Molecular sequences have been useful in identifying new taxa and providing preliminary hypotheses regarding the evolutionary history in this group. However, the lack of molecular sequences for the type species *Noblella peruviana* and *Psychrophrynella bagrecito* prevents a satisfactory resolution of this taxonomic issue. The 16S phylogeny we present here suggests that at least some species of *Psychrophrynella* may be nested within *Noblella*, assuming *N. thiuni* is closely related with the type species *N. peruviana*. Furthermore, our phylogeny shows that species of *Noblella* and *Psychrophrynella* from southern Peru form a distinct and not closely related clade with species of *Noblella* from northern Peru and Ecuador (*N. lochites* and *N. myrmecoides*), suggesting an alternative generic placement for these species. However, we refrain from making any taxonomic decision in absence of material from the type species of *Noblella* and *Psychrophrynella*.

## CONCLUSIONS

We describe a new species of terrestrial-breeding frog in the genus *Noblella*. We justify generic placement based on morphological similarity and phylogenetic analyses, but we note that such placement is tentative in light of the lack of known synapomorphies distinguishing *Noblella* from similar genera, particularly *Psychrophrynella*, and the absence of DNA sequence data for the type species of both *Noblella* and *Psychrophrynella*. We discuss limitations for singleton descriptions and adopt recent recommendations justifying our decision to proceed with a formal taxonomic decision for the new species. Our work contributes to documenting the rich diversity of small terrestrial-breeding frogs found at high elevations in the eastern slopes of the Andes of southern Peru and Bolivia.

## ACKNOWLEDGEMENTS

We thank A Shepack for lab assistance.

### Funding

This work was supported by grants from the Eppley Foundation, Wildlife Acoustics, and the Chicago Board of Trade Endangered Species Fund. The funders had no role in study design, data collection and analysis, decision to publish, or preparation of the manuscript.

### Grant Disclosures

The following grant information was disclosed by the authors:
The Eppley Foundation.
Wildlife Acoustics.
Chicago Board of Trade Endangered Species Fund.

### Competing Interests

The authors declare there are no competing interests.

### Author Contributions

- Alessandro Catenazzi conceived and designed the experiments, performed the experiments, analyzed the data, contributed reagents/materials/analysis tools, prepared figures and/or tables, authored or reviewed drafts of the paper, approved the final draft.
- Alex Ttito conceived and designed the experiments, performed the experiments, analyzed the data, contributed reagents/materials/analysis tools, authored or reviewed drafts of the paper, approved the final draft.

### Animal Ethics

The following information was supplied relating to ethical approvals (i.e., approving body and any reference numbers):

The Institutional Animal Care and Use Committee at Southern Illinois University Carbondale (SIUC IACUC protocol #16-006) and Florida International University (IACUC-18-009) approved this research.

### Field Study Permissions

The following information was supplied relating to field study approvals (i.e., approving body and any reference numbers):

Collecting permits were issued by the Dirección General Forestal y de Fauna Silvestre, Ministerio de Agricultura y Riego (N° 292-2014-MINAGRI-DGFFS-DGEFFS).

### DNA Deposition

The following information was supplied regarding the deposition of DNA sequences:

The 16S rRNA sequence is available at GenBank: accession number MK072732.

## Data Availability

Holotype (CORBIDI18723) is deposited at CORBIDI - Herpetology Collection, Centro de Ornitología y Biodiversidad, Lima, Peru. Photographs of live holotype are deposited in the online repository http://calphotos.berkeley.edu, automatically linked to AmphibiaWeb (http://amphibiaweb.org/).

## New Species Registration

The following information was supplied regarding the registration of a newly described species:

Publication LSID: urn:lsid:zoobank.org:pub:3F917F8B-5D31-44D4-86E2-389341A21BD1;

Noblella thiuni sp. n. LSID: urn:lsid:zoobank.org:act:1B9D35DE-0BA2-44E1-9559-62E30D239A30

## Supplemental Information

Supplemental information for this article can be found online at http://dx.doi.org/10.7717/peerj.6780#supplemental-information.

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
