# Peer review of "Noblella thiuni sp. n., a new (singleton) species of minute terrestrial-breeding frog (Amphibia, Anura, Strabomantidae) from the montane forest of the Amazonian Andes of Puno, Peru"

_PeerJ, doi:10.7717/peerj.6780_

## Round 0.1 · original submission · Major Revisions

Although both reviewers found merit in your paper, they also identified a number of issues that have to be addressed. I'd ask you to pay particular attention to the two main points raised by Reviewer 2 which must be adequately addressed.

·

Basic reporting

No comment.

Experimental design

No comment.

Validity of the findings

No comment.

Additional comments

The manuscript is solid and well-written. I only have one major comment. The species is described based on a single specimen; the authors should briefly discuss the problems associated with the low sample size of the description (e.g., sexual dimorphism, intraspecific variation). Also, I suggest adding a few sentences explaining why no additional specimens were included in the description... Is the species rare? Please describe the sampling effort at the type locality and nearby areas. Finally, add information about the conservation threats (if any) that the species might be facing.

·

Basic reporting

The authors describe a new species of Noblella from Southern Peru. The paper is well written and conforms to the “standard sections”. The authors used well referenced, actual and fundamental literature related to the subject. They present important new data on a very interesting and difficult group of direct developing frogs, however important information needed for the species description is limited or missing. These are discussed in the next section.

Experimental design

While the manuscript has the potential to be published, I think that there are a number of serious concerns that preclude its consideration for publication in its current state.

1. My main concern is that the description of the new species is based on only one specimen. While there are many cases when new species were described based on only one specimen it is important to address the problems associated to singleton descriptions (see Castroviejo-Fisher et al. 2011; Lim et al. 2012 and more recently Guayasamin et al. 2018). The authors fail to do this in the current manuscript. Almost no information concerning their field work is provided (when and how many times was the study site visited, methodology used etc.) and no details provided why they were not able to collect additional specimens (they just mention that the specimen was obtained during a “short survey” of the type locality - L 310). From personal experience I know how difficult it is to find the small frogs of the genus Noblella, but with regular visits (ideally in different seasons and weather conditions) is possible to collect a decent number of individuals needed for the description.

This problem is especially severe as the new species seems very similar to the problematic type species of Noblella, N. peruviana, for which we lack any new data (molecular and morphological) as no additional specimens were found since the original description. The morphological differences found between the new species and N. peruviana don’t seem to be sufficient to separate them, considering the scarce information: old specimens on one hand (possibly even from different species; see De La Riva et al. 2008) and only one specimen on the other. By the way, how far from the putative type locality of N. peruviana (the vicinity of the Inca Mine, in Santo Domingo de Carabaya, De La Riva et al. 2008) was collected the N. thiuni specimen? I strongly recommend that the authors address these issues in their discussion.

2. A second problem that I could identify is with the phylogenetic analysis. I understand that the authors didn’t try to resolve the phylogenetic relationships of this difficult group and their analysis is preliminary, but I still think that a robust analysis should be performed. I think that the used 16S sequences are not sufficient (especially for this group), and ideally inclusion of at least one nuclear gene would be desirable. Also, I would recommend the use of the primers 16SC and 16SD for the 16S gene as these are producing better quality sequences (more than 830 bp), in the case of Noblella, than the used 16Sar - 16Sbr pair. Additionally, I would recommend the use of a more versatile phylogenetic analysis tool like RAxML or Garli for the ML analysis (and with 1000 bootstrap replicates) and ideally coupled with a Bayesian inference in order to present a better resolved phylogenetic tree (as many groups had very low support in the authors Mega analysis). And finally and more importantly the authors don’t provide any information on the genetic distances of the closely related species. I recommend the inclusion of this data in the manuscript.

Validity of the findings

I’m afraid that the authors don’t provide sufficient data to support the validity of the new species in the manuscripts current form, especially because the analyzed specimen is so similar to a controversial species. I would recommend the inclusion of additional specimens in their description and the rerun of the molecular analysis.

Additional comments

Minor comments:

Figure 5 - Please correct the scale bar to mm: cannot be 2 cm!

Appendix 1 - Please add the legend for the collections abbreviations for the examined specimens

16S rRNA sequences (FASTA format) for the holotype of Noblella thiuni sp. n. - I don’t think this is useful as the sequence will be available from GenBank. It would be more useful (but not mandatory) to attach the whole alignment matrix.

Appendix 2 - is not properly presented as there is no legend and the listed species don’t match with the ones from the tree.

---

## Round 0.2 · Minor Revisions

Although the vast majority of the issues have been addressed, Reviewer 2 still identified a few details that need to be corrected.

·

Basic reporting

No comment

Experimental design

No comment

Validity of the findings

No comment

Additional comments

In my view, the authors have fully addressed the comments raised by myself and the other reviewers. I think that the description is solid and should be published.

·

Basic reporting

no comment

Experimental design

The authors added a detailed description of the problems related to singletons species and important data that was missing in the original MS. They also significantly improved the discussion section.

Validity of the findings

The authors managed to add sufficient information to support the validity of the new species. I think that the MS is publishable in its current form after correcting some very small errors presented in the next section.

Additional comments

Minor comments:

- I still would add in the Materials & methods section a short description of the field work: when and how the short survey was implemented and other details of the field work (how many persons, survey methods etc.). I would remove these details from the “Distribution, natural history and threats” and add them to the Materials & methods. Sorry, but I didn’t find sufficient information related to the field work not even in the article naming Psychrophrynella glauca. I think that it’s very important that you present the details of how your field work was carried out.

356 - “Santo Domingo is a cloud forest at 1690 m asl is in a small tributary of the upper Inambari River, ~150 km SE of the type locality of N. thiuni.” Something is missing from this sentence; please correct it.

Appendix 2 - Replace “Celio F. B. Haddad Field Series” with “CFBHT – Celio F. B. Haddad Field Series” (add CFBHT).

---

## Round 0.3 · accepted · Accept

I'm satisfied with the final modifications.

#